# Emergency Departments: An Underutilized Resource for Expanding COVID-19 Vaccine Coverage in Children

**DOI:** 10.3390/vaccines11091445

**Published:** 2023-09-01

**Authors:** Rebecca Hart, Yana Feygin, Theresa Kluthe, Katherine Quinn, Suchitra Rao, Shannon H. Baumer-Mouradian

**Affiliations:** 1Department of Pediatrics, Norton Children’s and the University of Louisville School of Medicine, Louisville, KY 40202, USA; 2Department of Psychiatry and Behavioral Medicine, Medical College of Wisconsin, Milwaukee, WI 53226, USA; 3Department of Pediatrics (Infectious Diseases and Hospital Medicine), University of Colorado School of Medicine and Children’s Hospital Colorado, Aurora, CO 80045, USA; 4Department of Pediatrics, Medical College of Wisconsin/Children’s Hospital of Wisconsin, Milwaukee, WI 53226, USA

**Keywords:** COVID-19, vaccination, pediatric, emergency department, healthcare disparities

## Abstract

COVID-19 vaccine (CV) acceptance rates remain suboptimal in children. Emergency departments (EDs) represent a unique opportunity to improve vaccination rates, particularly in underserved children. Little is known about the presence or reach of CV programs in US EDs. We assessed, via a cross-sectional survey of pediatric ED physicians, the number of EDs offering CVs to children, the approximate numbers of vaccines administered annually, and the perceived facilitators/barriers to vaccination. The proportion of EDs offering CVs is reported. Chi-square tests compared facilitators and barriers among frequent vaccinators (≥50 CVs/year), infrequent vaccinators (<50 CVs/year), and non-vaccinators. Among 492 physicians from 166 EDs, 142 responded (representing 61 (37.3%) EDs). Most EDs were in large, urban, academic, freestanding children’s hospitals. Only 11 EDs (18.0%) offer ≥1 CV/year, and only two (18.2%) of these gave ≥50 CVs. Common facilitators of vaccination included the electronic health record facilitation of vaccination, a strong provider/staff buy-in, storage/accessibility, and having a leadership team or champion. Barriers included patient/caregiver refusal, forgetting to offer vaccines, and, less commonly, a lack of buy-in/support and the inaccessibility of vaccines. Many (28/47, 59.6%) EDs expressed interest in establishing a CV program.

## 1. Introduction

COVID-19 infection remains a public health concern in the United States (US). While older patients account for the majority of hospitalizations and deaths, children represent more than 15% of total cases and can spread COVID-19 to both other children and adults [1,2,3,4,5]. As of June 2023, nearly 17 million pediatric cases of COVID-19 and 2267 deaths have been reported [6], with disproportionate numbers of both cases and deaths seen in children from racial and ethnic minority groups [7,8,9,10,11,12]. Decreasing infection rates in children is therefore imperative to avert the medical and social harms of COVID-19. Vaccination against COVID-19 mitigates illness severity, hospitalization, post-acute complications (long COVID) and Multisystem Inflammatory Syndrome in Children, and death, especially in those at a high risk for severe infection [13,14,15,16]. Unfortunately, despite the availability, safety, and efficacy of COVID-19 vaccines for children, pediatric vaccination rates remain suboptimal, especially among minority and underserved children [17,18,19,20,21,22]. Recent data from the Centers for Disease Control and Prevention (CDC) and the American Academy of Pediatrics (AAP) find that only 13% of children age 6 months to 4 years have received at least one dose of a COVID-19 vaccine, while 39% of 5–11-year-olds and 68% of 12–17-year-olds have had at least one dose [17]. While common reasons for low COVID-19 vaccination rates in children include parental hesitancy, concerns about safety/side effects, and perceptions that the vaccines are “too new” or that development was “rushed”, many parents also cite barriers such as a lack of transportation, a lack of medical insurance, or difficulty obtaining an appointment for their child to receive a CV [18,19,23,24,25,26,27].

The emergency department (ED) presents a unique opportunity to improve pediatric COVID-19 vaccination rates, especially in under-vaccinated children. The ED provides an opportunity for the administration of many vaccines, including influenza, and ED-based vaccinations have been demonstrated to be cost-effective and well-accepted by patients and providers [28,29,30,31,32,33,34,35]. EDs are also common sites of care for patients without a medical home and without insurance, as well as those with complex medical needs who may be at increased risk of COVID-19 illness [36,37,38]. The ED also serves as a convenient site for vaccination, overcoming commonly cited barriers such as access to transportation, difficulty making an appointment, and insurance problems [39,40]. A recent study of 22 pediatric EDs demonstrated that Black (40%) and Hispanic (17%) children, and those with public health insurance (51%), are the primary utilizers of pediatric EDs and are more likely to have recurrent ED visits [41]; therefore, ED-based COVID-19 vaccine (CV) programs may be important strategies to increase vaccination rates among these children. Further, our prior research has demonstrated that caregivers are willing to receive COVID-19 vaccines for their children in the pediatric ED, and find the ED a convenient site to receive CVs [26,42].

The literature related to pediatric ED vaccination programs for COVID-19 is limited. Therefore, we sought to assess the current presence and reach of pediatric ED-based COVID-19 vaccine programs, the characteristics of these programs, and the perceived facilitators and barriers to such programs. We hypothesized that fewer than 10% of pediatric EDs currently offer CVs, and that common barriers to offering CVs would include a perceived lack of patient interest and a lack of resources/staffing to provide vaccines, while common facilitators would include a strong buy-in from nurses and providers and a vaccine leadership team or champion. 

## 2. Materials and Methods

We conducted a cross-sectional survey of pediatric ED physicians, which was reviewed and approved for distribution by the Pediatric Emergency Medicine Collaborative Research Committee (PEM CRC), a group of pediatric emergency medicine physicians representing up to 166 EDs who are members of the American Academy of Pediatrics (AAP) Section on Emergency Medicine. Distribution of a survey through the PEM CRC is a competitive process. The PEM CRC committee reviews surveys on a rolling basis and selects one survey to be distributed to its members quarterly. 

Our study aims were to establish the proportion of responding EDs who administered any CV in the past year, and to categorize the number of vaccines administered annually (<1—non-vaccinating, 1–50—infrequent vaccinator, 51 or more—frequent vaccinator). We also sought to compare ED-based characteristics among vaccinating and non-vaccinating EDs and to determine and compare perceived facilitators and barriers to CV programs among vaccinating and non-vaccinating EDs. We utilized an arbitrary and conservative estimate of >50 vaccines administered annually to describe “frequent vaccinator” programs. No existing literature identifies a number of CVs that establishes clear “success” in this setting. The authors determined from prior experience with influenza vaccine programs that vaccinating anywhere from 5 to 10% of eligible patients is rare and seen only among the most successful ED influenza vaccine programs. Therefore, we assumed that EDs administering <50 COVID-19 vaccines per year would be significantly below this 5% margin.

The primary outcome was the proportion of unique EDs with an existing CV program (administered any CV per year). The secondary outcomes included the number of CVs administered per year (allowing us to categorize them as frequent or infrequent vaccinators based on the number of vaccines administered), and the facilitators and barriers to vaccination. 

### 2.1. Survey Development and Distribution

We designed the survey questions for our specific setting and population by adapting similar questions from the literature; additional novel questions were developed to address the specific aims for our study [35,43,44,45]. Questions included information about the respondent and their primary ED (including the respondent’s role, the ED’s annual volume, location, and the proportion of patients receiving government insurance), whether the ED offers CVs, and how many vaccines were administered in the past year. Facilitators and barriers were assessed via the question: “Which of the following do you perceive to be [facilitators/barriers] of COVID-19 vaccination to children in your ED?” using a 5-point Likert scale and a list of potential facilitators and barriers, which were selected based on the prior literature as well as a priori by the investigators and their prior research in this area [26,35,39,46]. Additional questions assessed (1) the desire to collaborate to start or improve a CV program and (2) the desire to participate in future focus groups related to barriers and facilitators of vaccine programs. 

Content experts on vaccination, statistical analysis, and PEM survey design systematically reviewed the survey for clarity and relevance to ensure that individual survey items were appropriate, relative to the construct being measured, and to improve overall quality. Cognitive interviews were conducted with nine PEM faculties to ensure an interpretation of the questions and the answer stems in the manner intended by the investigators, and to ensure that PEM providers had adequate knowledge of their ED’s vaccine practices to address the questions.

The survey was transcribed into the Qualtrics XM survey management software and piloted with thirteen additional PEM providers representing six unique sites within the COVID-19 Parent Attitudes Study (COVIPAS) group. Multiple providers at each site completed a pilot test of the survey to assess for the time required to complete the survey and the understanding of the questions as designed, and to allow investigators to determine whether responses were reliable among multiple respondents from a single institution. Pilot responses were not included in the final analysis.

The PEM CRC committee reviewed the final survey, and further revisions were made to improve clarity. The survey was distributed electronically to the 492 members of the PEM CRC, representing up to 166 EDs across the US, for completion between 1 October and 6 December 2022. Three email reminders to complete the survey were sent before 1 December 2022. The study was approved by the Institutional Review Board at the University of Louisville.

### 2.2. Analysis

Descriptive statistics assessed the demographic questions, which are reported by program. The proportion (with 95% confidence interval) of EDs with an active CV program was reported overall and by performance category, defined as the number of vaccines administered. If there were inconsistencies in responses on any key survey question (type of vaccine available, number of vaccines administered, demographics of vaccinated population), the primary author contacted the division/section chief or medical director at that institution for final adjudication. Likert scores for the perceived barriers/facilitators were assessed by the individual respondents and were dichotomized into “agree” (Likert score 4–5, agree or strongly agree) or “disagree” (Likert score 1–3, strongly disagree, disagree, or neutral). Chi-square tests compared the most common perceived barriers between EDs with and without CV programs (non-vaccinators vs. frequent/infrequent vaccinators combined), as well as between frequent and infrequent vaccinator EDs. Odds ratios (OR) were reported with 95% confidence intervals (CI). Analyses were conducted using R statistical software, version 4.2.1 (R Foundation for Statistical Computing, Vienna, Austria).

## 3. Results

Surveys were emailed to 492 physicians representing up to 166 unique EDs (total number estimated as some respondents declined to document their facility), and 142 (28.9%) of these physicians responded, representing 61 (37.3%) unique EDs. Nearly all (58/61, 95.1%) programs showed an agreement between individuals from the same institution related to the number and type of vaccines offered; three programs had discrepant answers, which were all clarified through email contact with the leadership at the individual institutions. The surveyed EDs were primarily academic, freestanding children’s hospitals in urban settings, with annual volumes >30,000 visits per year (Table 1). Almost half (N = 28, 48.9%) of the programs cared for a predominantly publicly insured population (>60% patients with public insurance). 

For the primary outcome, we found that 11/61 (18.0%) of the responding EDs currently offered COVID-19 vaccines. Of these 11 EDs, most (9/11, 81.8%) had administered <50 CVs in the past year (infrequent vaccinators), two (18.2%) had administered 50–999 CVs (frequent vaccinators), and none had administered >1000 CVs. Among EDs with CV programs, three (27.3%) had an established workflow for offering CVs, two of whom were frequent vaccinators. Frequent vaccinators were significantly more likely to have a tailored vaccine workflow than infrequent vaccinators (*p* = 0.003). Further differences between the demographics of frequent and infrequent vaccinators and non-vaccinating programs are summarized in Table 1. Most notably, the two frequent vaccinators were both urban, academic, freestanding children’s hospitals with >30,000 patient visits per year. The infrequent vaccinators and non-vaccinators had generally similar demographics, although these were also primarily urban, academic, freestanding children’s hospitals; infrequent vaccinators were more likely to see >60% of patients with public insurance compared to non-vaccinators, frequent vaccinators, or the overall respondent pool. 

Figure 1 demonstrates the proportion of respondents who agreed or strongly agreed that certain factors were facilitators of CVs in their ED. Among EDs that offer CVs, commonly noted facilitators of CV programs included electronic health record (EHR) tools to facilitate vaccination (78.6% rated agree or strongly agree), the presence of a vaccine leadership team/champion (71.4%), the Vaccine for Children (VFC) program facilitating payment for vaccines (71.4%), vaccines being readily accessible/stored in the ED (64.3%), and staff buy-in (60.0%). Notably, 64.3% of respondents from an ED that currently offers CVs felt that their program was not yet successful at offering and administering CVs. 

Figure 2 demonstrates the proportion of respondents who agreed/strongly agreed that certain factors were barriers to administering CVs. Among EDs that offer CVs, common barriers to vaccination included high patient refusal rates (60.0% agree or strongly agree) and forgetting to offer vaccines (90.0%). Less common barriers for this group included a lack of buy-in from staff (30.0%), a lack of administrative support (30.0%), vaccines not being easily accessible in the ED (30.0%), and a lack of vaccine leadership (20.0%). Among non-vaccinating EDs, respondents indicated that commonly perceived barriers included patient refusal rates (75.0% agree or strongly agree), forgetting to offer vaccines (87.5%), or EDs having other priorities precluding COVID-19 vaccination efforts (44.4%). A lack of buy-in from staff was also more frequently cited by non-vaccinating EDs (44.4%) compared to those offering CVs. Notably, EDs that were frequent vaccinators were generally less likely to identify each of the above categories as barriers than infrequent or non-vaccinators. 

Notably, respondents from 28/47 (59.6%) of the non-vaccinating EDs expressed interest in establishing a CV program, 20 (71.4%) of whom provided contact information for a future working collaboration or ongoing study. 

## 4. Discussion

In this national survey of pediatric ED providers, we found that less than one quarter of EDs offer CVs to children, with very few providing >50 vaccines annually. This implies a significant missed opportunity for targeting vaccination efforts, particularly for underserved and high-risk patients who are frequent utilizers of ED care. Facilitators of vaccination included CV-specific EHR tools, the presence of a vaccine leadership team or champion, VFC program payment for vaccines, an easy accessibility/storage of vaccines, and a strong buy-in from providers and nurses. Barriers to vaccination included patient refusal, forgetting to offer vaccines, and (less commonly) a lack of prioritization or lack of buy-in from staff/administration. 

To our knowledge, this is the first study of its kind to date to assess the availability of CVs for children in US EDs, and to explore facilitators/barriers experienced by pediatric EDs when implementing CV programs. Facilitators and barriers to CV administration have been widely discussed in the literature from the patient and caregiver perspective, with frequent mentions of issues with healthcare access [18,19,20,25,26,27,47,48,49]. For this reason, exploring opportunities to vaccinate in the ED is an important area of study. Prior studies note that many parents who intend to vaccinate their children against COVID-19 would accept a CV in the ED setting, with convenience being a key factor in that decision [26,42]. As a result, the CDC has made recommendations that CVs be available to patients at time of hospital discharge or during ED visits [50]. The American College of Emergency Physicians (ACEP) has agreed that EDs represent a potentially important public health opportunity for COVID-19 vaccination programs and has developed a toolkit for ED CV programs in general/adult EDs [51]. However, no such toolkit has yet been devised for pediatric EDs or specifically related to pediatric COVID-19 vaccination. As many respondents in this study indicated interest in establishing a CV program, the development of a pediatric-specific “toolkit” that incorporates these strategies may be of benefit in the future. 

Unfortunately, little is known about the prevalence of CV programs in the pediatric ED, or about the barriers and facilitators to the implementation of such programs from the ED provider/facility perspective. One study of ED and nursing department heads in French EDs demonstrated that perceived factors limiting the ability of EDs to vaccinate patients against COVID-19 included overcrowding, a lack of medical staff, and a lack of patient follow-up [35]. Other studies have shown that CV programs are feasible and can increase the vaccination of vulnerable groups in adult populations in both the US and Australia [52,53]. We expand on this literature by assessing the practical experiences of pediatric EDs related to CV programs for children in the US. The common facilitators we identified highlight the need for programs that wish to begin offering CVs to solicit support from their providers and nursing staff, to establish a leadership team or “champion(s)” who can lead implementation efforts, to establish a workflow for regularly screening patients (particularly by utilizing tools in the EHR), and to ensure the accessibility/storage of CVs in the ED itself. While caregiver/patient refusal was a commonly perceived barrier among non-vaccinating EDs, it was less frequently experienced by frequent vaccinators than infrequent vaccinators. This may be due to the presence of workflows, processes, and communication strategies that improve the discussion and buy-in from patients among these frequent vaccinators, and should be explored further. As even frequent vaccinators often agreed that patient/caregiver refusal was a barrier to vaccination, targeted strategies to discuss and overcome vaccine hesitancy will be important to develop in future studies. 

COVID-19 vaccines are complex to administer, particularly compared with influenza, tetanus, and other vaccines that are commonly utilized in the ED setting. Frequent changes in the recommendations related to the number of doses, required boosters, and the composition/included strains of vaccines, along with requirements for cold storage, drawing from multi-dose vials while minimizing waste, and the need to arrange for follow-up doses at various timepoints, all contribute to the unique challenges of offering and administering CVs [54]. This study confirms that ED providers identify ease of storage and access to CVs as facilitators of vaccination, although we did not expand on additional structural or logistical concerns that may exist. Hospital-based CV programs have been identified (and recommended) as a promising route to reach populations with decreased vaccine access and higher risk from COVID-19 infection, as hospitals typically have the storage and infrastructure capabilities to overcome these challenges [50,54]. EDs may have similar capabilities, but identifying feasible ways to overcome the unique challenges of CV administration specifically will be key for promoting and sustaining CV programs in this setting. Further, processes to appropriately refer patients to appropriate centers (e.g., primary care providers or pharmacies) for the additional required doses of the CV will be the key to successful ED campaigns. Some non-vaccinating EDs expressed concern about these logistics as a barrier to vaccination; however, this was generally not a barrier to frequent vaccinators. Further detail about the methods for referral and ongoing care utilized by frequent vaccinators is needed in order to guide EDs that wish to initiate and expand CV programs. 

While this study did not specifically inquire about racial or ethnic identity among ED patients, the prior literature identifies that patients who identify as ethnic or racial minorities are disproportionately likely to utilize pediatric EDs [41], as are children who are under/uninsured, who have public health insurance, who are without a medical home, or who have complex medical needs [36,55]. There is a significant overlap between this population of frequent ED users and children who are at the highest risk of complications of COVID-19 [7,8,9,10]. Given the disproportionate utilization of EDs by these populations, the provision of CVs in the ED may help to decrease the burden of COVID-19 illness among minority and high-risk patients. The investigators wish to highlight the insurmountable evidence that race is not a biologic proxy; as such, we acknowledge that differences in these outcomes are representative of systemic, cultural, environmental, and socioeconomic differences in the lived experiences and exposures of these populations. The authors have previously published work demonstrating that the convenience of ED vaccination programs is a potential facilitator of CV administration among Black parents [26], further emphasizing the need for the expansion of ED CV programs. Further multicenter studies are needed to evaluate the impact of CV programs on CV acceptance in racial and ethnic minorities; establishing specific strategies related to these populations will be critical to the development of future guidelines or tools to establish and grow ED CV programs. 

While this study represents the views of providers from across the US, we received responses from less than half of EDs within the PEM CRC network; responses may therefore not fully represent the CV programs in EDs nationally. Specifically, the use of this network may bias the responses toward opinions representing academic freestanding children’s hospitals. Respondents’ individual beliefs and practices may have influenced their responses and may not reflect the experience or opinions of their colleagues in the ED. However, the broad representation of EDs from varying locations, sizes, and affiliations implies an adequate representation of a variety of viewpoints. Additionally, the interest in a collaboration and expansion of CV programs by participants demonstrates that this is a common interest among many facilities, and even if results were affected by response bias, the future impact on children’s health by implementing CV programs among interested facilities remains significant. We did not inquire about specific practices utilized by EDs who offer CVs, such as methods of screening or delivery, nor about how EDs arrange for additional doses of CVs for patients completing the primary series. These specifics may help to guide other EDs who are interested in developing CV programs, and establishing further details on recommended practices will be necessary in future studies. Finally, this study represents data collected over a single time point (fall 2022); as such, more EDs may have adopted CV programs since the time of the survey. 

## 5. Conclusions

Less than one-quarter of participating pediatric EDs currently offer CVs to children. Leveraging the identified facilitators and overcoming barriers to develop, implement, and/or improve CV programs in EDs may contribute to improving COVID-19 vaccination rates among children. Children from racial/ethnic minority backgrounds, those who are underserved, and those at high risk for complications from COVID-19 may particularly benefit from such programs. These data justify the need to develop a toolkit and to model workflow for the administration of COVID-19 vaccines in the ED that addresses missed opportunities for vaccinating children. The development of infrastructure and support systems for CV administration may lead to future support of additional vaccination efforts for other pandemic and/or endemic infectious diseases (such as influenza, RSV, and others) in the future. 

## Figures and Tables

**Figure 1 vaccines-11-01445-f001:**
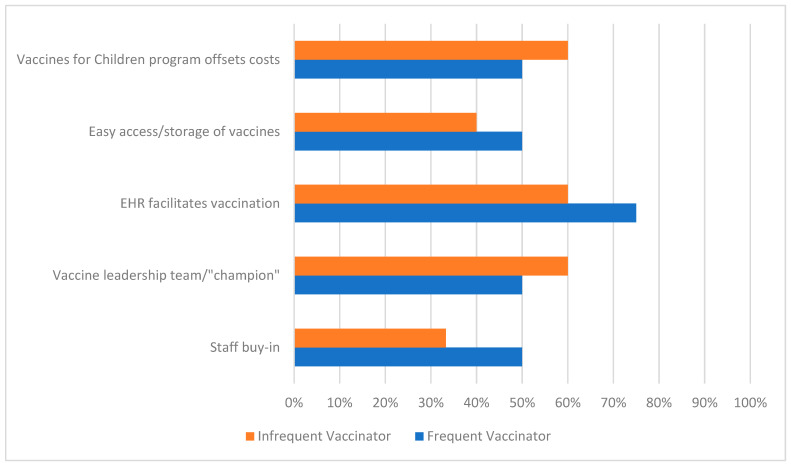
Proportion of frequent vs. infrequent vaccinators who agree/strongly agree that factors are facilitators of COVID-19 vaccination in the ED.

**Figure 2 vaccines-11-01445-f002:**
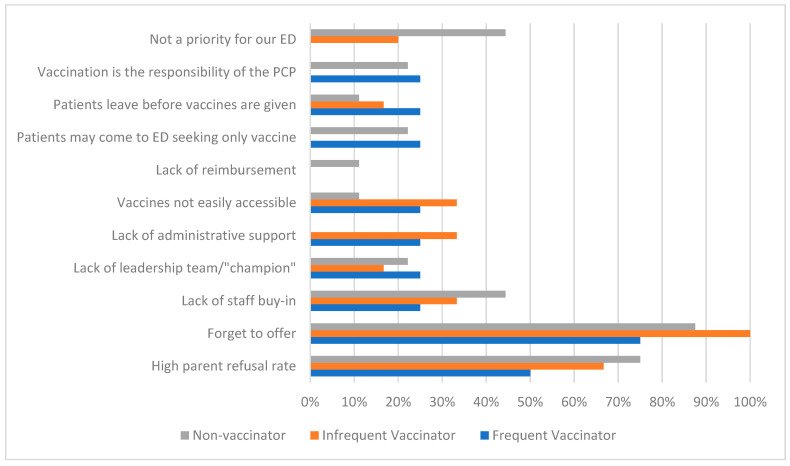
Proportion of frequent vaccinators, infrequent vaccinators, and non-vaccinators who agree/strongly agree that factors are barriers to COVID-19 vaccination in EDs.

**Table 1 vaccines-11-01445-t001:** Responding ED demographics, overall and by vaccination program status.

Factor	Overall N = 61 N (%)	Frequent Vaccinators N = 2 N (%)	Infrequent Vaccinators N = 9 N (%)	Non- Vaccinators N = 50 N (%)	*p*-Value
**ED Facility**					
Freestanding children’s hospital	33 (54.1)	2 (100.0)	5 (55.6)	26 (52.0)	0.749
Dedicated pediatric ED/hospital within an adult facility	24 (39.3)	0 (0.0)	4 (44.4)	20 (40.0)	
Primary adult facility that serves children	4 (6.6)	0 (0.0)	0 (0.0)	4 (8.0)	
**ED Setting**					
Urban	49 (80.3)	2 (100.0)	7 (77.8)	40 (80.0)	>0.999
Suburban	11 (18.0)	0 (0.0)	2 (22.2)	9 (18.0)	
Rural	1 (1.6)	0 (0.0)	0 (0.0)	1 (2.0)	
**Academic Classification**					
Academic	53 (86.9)	2 (100.0)	9 (100.0)	42 (84.0)	0.809
Non-academic	5 (8.2)	0 (0.0)	0 (0.0)	5 (10.0)	
Other	3 (4.9)	0 (0.0)	0 (0.0)	3 (6.0)	
**Annual Patient Volume (visits/year)**					
60,000+	23 (37.7)	1 (50.0)	3 (33.3)	19 (38.0)	>0.999
30,000–59,999	20 (32.8)	1 (50.0)	3 (33.3)	16 (32.0)	
<30,000	18 (29.5)	0 (0.0)	3 (33.3)	15 (30.0)	
**Public Insurance (proportion of patients)**					
>60%	28 (45.9)	0 (0.0)	7 (77.8)	21 (44.7)	0.183
30–60%	28 (45.9)	2 (100.0)	2 (22.2)	24 (51.1)	
<30%	2 (3.3)	0 (0.0)	0 (0.0)	2 (4.3)	
No answer/not applicable	3 (4.9)	0 (0.0)	0 (0.0)	0 (0.0)	
**Electronic Health Record**					
EPIC	46 (75.4)	0 (0.0)	8 (88.9)	38 (76.0)	0.185
Cerner	12 (19.7)	2 (100.0)	1 (11.1)	9 (18.0)	
Allscripts	2 (3.3)	0 (0.0)	0 (0.0)	2 (4.0)	
Other	1 (1.6)	0 (0.0)	0 (0.0)	1 (2.0)	
**COVID-19 Vaccine (CV) Available**	14 (23.0)	2 (100.0)	9 (100.0)	0 (0.0)	<0.001
**Standardized Workflow for ** **Offering CVs**	3 (4.9)	2 (100.0)	1 (12.5)	0 (0.0)	0.003

## Data Availability

Data are available by contacting the corresponding author, Rebecca Hart MD MSc, at becca.hart@louisville.edu.

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
