# Peer review of "Emergency Departments: An Underutilized Resource for Expanding COVID-19 Vaccine Coverage in Children"

_vaccines, 2023, doi:10.3390/vaccines11091445_

Round 1

Reviewer 1 Report

Thank you for your submission on this interesting topic. I have a few suggestions for your review.

Introduction: Page 2 line 46 have a closed parentheses that does not need to be there.  Line 57 review sentence should that read "important" ?  It may be helpful to describe how EDs that administered COVID vaccines handled the additional doses needed to complete the series.

Methods: It would be good for the reader to know in the methods which areas (rural, urban etc) where the surveys were sent- it was not mentioned until the results.  The reader needs to understand the exact methods.  Did you sent the survey to multiple states or just one state?  Define exactly what 'Specific settings" were.

 Discussion: Addressed the objectives and limitations. 

References:

Reference #2: Should that read "Children" ?

AMA Format used abbreviations for journals not  the full title of the journal. Review: references:  6,10, 32, 35, 36, 48

Is Reference 34 a published abstract?

Reference 49- please provide date accessed

Grammar: Please check spacing throughout the manuscript- there many places that have double space between words

Page 8 line 234 and page 9 line 277- have double commas

Grammar: Please check spacing throughout the manuscript- there many places that have double space between words

Page 2 line 46 have a closed parentheses that does not need to be there.  Line 57 review sentence should that read "important" ?

Page 8 line 234 and page 9 line 277- have double commas

Reference #2: Should that read "Children" ?

Author Response

Point 1: Page 2 line 46 have a closed parentheses that does not need to be there.  

Response 1: This has been corrected.

Point 2: Line 57 review sentence should that read "important" ?  

Response 2: This has been corrected to "important." 

Point 3: It may be helpful to describe how EDs that administered COVID vaccines handled the additional doses needed to complete the series.  

Response 3: We agree, however, the survey as delivered unfortunately did not inquire about specific details related to delivery of vaccines nor setup of additional doses or appointments that might be necessary for patients completing the primary series. We believe this is an area that can and should be addressed in future studies. We have added to the discussion, lines 327-332, to address this specifically: "We did not inquire about specific practices utilized by EDs who offer CV, such as methods of screening or delivery, nor about how EDs arrange for additional doses of CV for patients completing the primary series. These specifics may help guide other EDs who are interested in developing CV programs, and establishing further details on recommended practices will be necessary in future studies." This topic is further addressed (in less detail) in the discussion, lines 308-316 and 320-325. 

Point 4: It would be good for the reader to know in the methods which areas (rural, urban etc) where the surveys were sent- it was not mentioned until the results.  The reader needs to understand the exact methods.  Did you sent the survey to multiple states or just one state?  Define exactly what 'Specific settings" were.

Response 4: We have additionally clarified the distribution in the methods, lines 133-134: The survey was distributed electronically to the 492 members of the PEM CRC, representing up to 166 EDs across the US, for completion between October 1st and December 6th, 2022.

Point 5: Reference #2: Should that read "Children" ? 

Response 5: This has been corrected to appropriately read "Children." 

Point 6: AMA Format used abbreviations for journals not  the full title of the journal. Review: references:  6,10, 32, 35, 36, 48

Response 6: These abbreviations have been corrected.

Point 7: Is Reference 34 a published abstract? 

Response 7: This references an abstract presented at the AAP NCE in October 2021 but was not published; the data has been submitted but not yet accepted for publication. 

Point 8: Reference 49- please provide date accessed 

Response 8: Date accessed has been added. 

Point 9: Please check spacing throughout the manuscript- there many places that have double space between words 

Response 9: Spacing has been corrected throughout manuscript.

Point 10: Page 8 line 234 and page 9 line 277- have double commas 

Response 10: These have been corrected/deleted.

Reviewer 2 Report

High quality manuscript describing the survey of covid vaccination programs in pediatric emergency department in the United States. No major concerns with the content or presentation.

Authors may wish to expand the introduction to emphasize their motivations for focusing on pediatrics. The burden of Covid-19 as an epidemic disease was principally in older adults and those with comorbidities, with lower impact on children. This may change as the disease transitions to endemic or seasonal because children <5yrs would be the dominant population of naïve individuals.

Author Response

Point 1: Authors may wish to expand the introduction to emphasize their motivations for focusing on pediatrics. The burden of Covid-19 as an epidemic disease was principally in older adults and those with comorbidities, with lower impact on children. This may change as the disease transitions to endemic or seasonal because children <5yrs would be the dominant population of naïve individuals.

Response 1: We agree that the importance of pediatrics in the focus of this paper was understated. We have expanded the initial paragraph to focus on the importance of children as a vector for disease in the pandemic, as well as emphasizing the potential economic and social burden that pediatric COVID-19 illness places on patients, families, and communities. Please see changes in paragraph 1, lines 31-34 and 36-37. 

Reviewer 3 Report

Introduction

Please include the common reasons of low CV in children in US. For example, are parents afraid of ADR associated with CV or any others. Also discuss how CV availability will overcome low CV rates among this population. Authors are trying to convey that establishment of CV centers in the EDs will enhance CV rates but please build rational of the study by including common factors associated with low CV rates in children.

EDs are usually are quite busy place in a hospital setting along with overflow of the patients all the time. So how EDs can manage CV in a place where working staffs already facing challenging environment?. 

Methods

It would be good if author can provide statics about the current influenza vaccines rates in EDs.

Please add the duration of data collection from the potential participants.

Minor editing of English language required.

Author Response

Point 1: Please include the common reasons of low CV in children in US. For example, are parents afraid of ADR associated with CV or any others. Also discuss how CV availability will overcome low CV rates among this population. Authors are trying to convey that establishment of CV centers in the EDs will enhance CV rates but please build rational of the study by including common factors associated with low CV rates in children.

Response 1: 

This discussion has been added to the introduction, lines 47-51. The importance of the ED in overcoming common factors centers on the ability to reach patients for whom hesitancy is less of an issue, but who may remain unvaccinated due to lack of access to primary care, lack of transportation, or other barriers. This is hopefully better clarified with the addition of the paragraph above, which links to further discussion in introduction lines 59-61. Additionally, in the discussion lines 274-275 we note that CDC recommendations encourage CV at any timepoint during which pts might present for care, including during ED visits.

Point 2: EDs are usually are quite busy place in a hospital setting along with overflow of the patients all the time. So how EDs can manage CV in a place where working staffs already facing challenging environment?. 

Response 2: This is certainly a concern and potential issue for the development of resilient ED vaccine programs. The specific logistics of how successful programs function in terms of screening, storing, and administering vaccines in the busy ED environment were not explored in this study. While the authors have personal experience from our own institutions, we felt this was beyond the scope of this paper. We have explored the potential logistical complexities of offering CV in the discussion, lines 308-325 and lines 358-363, and believe this is an excellent area for future studies/next steps to explore the more granular processes that EDs use and how they may overcome these environmental challenges.

Point 3: It would be good if author can provide statics about the current influenza vaccines rates in EDs.

Response 3: 

Could the reviewer clarify—do you mean the current number/proportion of EDs that offer influenza? Or acceptance rates of influenza vaccines in the ED? Literature (including studies published from the authors’ home institutions) suggests that acceptance rates for influenza vaccine in the peds ED may range from 30-55% among eligible patients. Unpublished data from the authors’ institutions which was obtained very early in the pandemic, when overall hesitancy was higher and vaccination rates lower, also suggests that COVID-19 vaccine uptake is likely lower (in the 20-30% range), but is quite high (up to 70-80%) among parents who intend to vaccinate their children. We are unsure if this data on influenza vaccines is relevant to this particular study, but we are happy to add it if the publishers feel this would improve the quality of the paper.

Point 4: Please add the duration of data collection from the potential participants.

Response 4: This is addressed in lines 155-157—emails were initially sent on 10/1/2022 and the survey was closed on 12/6/22. Three reminder emails were sent between 10/1 and 12/1.

Reviewer 4 Report

This is a very interesting piece of work. I find it quite insightful and a promising strategy that could be used to improve uptake of CV vaccines. It would be helpful if the authors can add a diagram that depicts the workflow for CV in ED. This might help other researchers or public health systems. 

I find the paper well-written and do not have further comments.

Author Response

Point 1: It would be helpful if the authors can add a diagram that depicts the workflow for CV in ED. This might help other researchers or public health systems. 

Response 1: 

Unfortunately, we did not inquire about specific possible workflows or recommended practices used by individual EDs, but have added to the discussion, lines 327-332 (quoted below), to address this fact and to emphasize that further detail through additional study is warranted.

"We did not inquire about specific practices utilized by EDs who offer CV, such as methods of screening or delivery, nor about how EDs arrange for additional doses of CV for patients completing the primary series. These specifics may help guide other EDs who are interested in developing CV programs, and establishing further details on recommended practices will be necessary in future studies."

Reviewer 5 Report

Although the goal of providing COVID-19 coverage to children presenting to the ER for whatever reasons is important, there are several complications/issues that need to be further addressed and discussed, eg: -parental consent -referral to primary care or commercial pharmacies for subsequent boosters, or to the authors think it reasonable that these children will return to the ER for booster doses? -is there any intention to provide vaccines for other infections? -is there any intent to provide further care for these children? As for my recommendation to editors, I think the authors have oversimplified the recommendation to provide COVID-19 vaccines to children who present to the ER for other reasons, with the exception of Childrens Hospitals, and even then, it is unlikely that this recommendation will result in any important change in combating the COVID-19 infection. Thank you for the opportunity to review this work.

None other than those given above.

Author Response

Although the goal of providing COVID-19 coverage to children presenting to the ER for whatever reasons is important, there are several complications/issues that need to be further addressed and discussed, eg:

-parental consent

We did not specifically inquire about consent practices used by EDs for vaccinations in this study. Anecdotally, consent practices would likely be similar to those utilized for other vaccines (such as tetanus) provided in the ED setting—however, this would be an area that would be useful to expand upon in future planned studies that will delineate recommended best practices for CV administration.

-referral to primary care or commercial pharmacies for subsequent boosters, or to the authors think it reasonable that these children will return to the ER for booster doses?

Please see the discussion section, lines 281-301. We had previously expanded on the complexity of CV administration that may provide challenges in the ED, and have expanded on this issue specifically relating to the need for subsequent boosters and referral care.

-is there any intention to provide vaccines for other infections?

While not within the scope of this article, the full survey that we performed did also ask about influenza vaccine programs in the ED. This work has been submitted for publication and is pending. We do foresee that the development and improvement of infrastructure for CV administration will be critical for other future vaccination efforts. If facilities are prepared with workflows, screening processes, communication strategies, etc. for COVID-19 vaccines, these practices will likely transfer easily to practices related to pandemic influenza, RSV, or other new vaccines that may arise in the future. We have added a line to the conclusion to this effect: “Development of infrastructure and support systems for CV administration may lead to future support of additional vaccination efforts for other pandemic and/or endemic infectious diseases (such as influenza, RSV, and others) in the future.” (Conclusions, lines 351-354).

-is there any intent to provide further care for these children?

We did not specifically inquire about further care for children who have received COVID-19 vaccines in the ED. This would be an excellent area to expand upon in our planned future studies.

Thank you for the opportunity to review this work.